# Exon sequencing mutation detection algorithm based on PCR matching

**Guobin Chen**[1,2], **Xianzhong Xie**[1]*

**1** College of Computer Science and Technology, Chongqing University of Posts and Telecommunications, Chongqing, PR China, **2** Chongqing Key Laboratory of Spatial Data Mining and Big Data Integration for Ecology and Environment, Rongzhi College of Chongqing Technology and Business University, Chongqing, PR China

* xiexzh@cqupt.edu.cn

## Abstract

### Background

With the development of second-generation sequencing technology, more and more DNA sequence variations have been detected. Exon sequencing is the first choice for sequencing many cancer genes, and it can be better used to identify disease status by detecting gene variants. PCR sequence is an effective method to capture that sequence of an exon in the process of sequencing. Exon sequencing sequence contains PCR primer sequence, the correct position of the sequence can be determined by PCR primer sequence, which can be found in SNP, Indel mutation point by comparing the sequence of PCR primer sequence.

### Results

In this paper, a matching algorithm based on the PCR primer sequence is proposed, which can effectively sequence the position of PCR primer sequence and find out the key position sequence. Then the sequencing sequence is sorted and the number of the same sequence is counted to reduce the matching times. Then, the sequenced sequence was matched with PCR primer sequence, so that the DNA position could be accurately matched and the variation in the sequenced sequence could be found more quickly.

### Conclusions

Compared with the traditional sequence matching method, PCR primer sequence matching method can match many sequences and find more variation. It also showed a high recall rate in the recall rate.

## Introduction

With the gradual maturation of the second-generation sequencing technologies [1], higher and higher sequencing throughputs and cheaper and cheaper sequencing [2], sequencing technologies are increasingly used in the diagnosis of disease genetics [3]. Many disease-related

**Data Availability Statement:** All data are fully available without restriction. The data can be accessed at https://doi.org/10.6084/m9.figshare.12660500.v1.

**Funding:** This work supported by the National Nature Science Foundation of China (Grants

No.61271259 and 61601070), the Chongqing Nature Science Foundation(Grants No. CTSC2011jjA40006, CSTC2010BB2415 and CSTC2016jcyjA0455), the Research Project of Chongqing Education Commission(Grants No. KJ120501, KJ12050, KJ1600411, KJ110530); The Key Project of Science and Technology Research of Chongqing Education Commission (KJZD-K201800603, KJZD-M201900602); Doctoral high school talent training project (BYJS2016003), Chongqing Graduate Scientific Research Innovation Project (CYB17131). The funding body didn't play any roles in the design of the study and collection, analysis, and interpretation of data and in writing the manuscript; the Science and Technology Research Program of Chongqing Municipal Education Commission (Grant No.KJZD-K201902101);Humanities and Social Sciences Project of Rongzhi College of Chongqing Technology and Business University(Grant No.20197004).

**Competing interests:** The authors have declared that no competing interests exist.

SNPs [4] have been studied more and more by sequencing techniques. In clinical medicine, the diagnosis of certain diseases and the search of target sites for the action of certain drugs involve only part of the genetic locus [5]. Although whole genome re-sequencing can also achieve these functions [6], Genome re-sequencing will detect those unrelated gene sequences together, which will result in a lot of financial and time waste, but also increase the difficulty of post-processing sequencing data. Under the premise that related reference genes corresponding to some traits of a species are known, some samples do not require genome-wide sequencing when performing sequence analysis on certain specific fragments in the gene. Only by sequencing the target region of specific interest in the sample gene will it be possible to know whether the sample has the potential to express the gene at the gene level. This alternative to genome-wide re-sequencing, which only targets the segments, not only greatly reduces sequencing costs, but also reduces subsequent data processing efforts.

Targeted sequencing [7] is a technique for sequencing common exons [8] in specific regions of genes in DNA sequences. This sequencing method is based on the susceptibility genes and can be used to determine whether a certain disease Disease), the region of the relevant gene is detected to detect whether there is a malignant mutation, and if so, the disease is determined.

Due to the short sequence length measured by the sequencer, in order to be able to completely sequenced the DNA sequences in the class, the DNA genome was randomly interrupted by ultrasonic waves into small fragments, and the fragments were added to both ends of the small fragments by PCR By technique [9], these small fragments are sequenced by the sequencer. However, this sequencing requires a certain depth of sequencing in order to meet the corresponding coverage. The deeper the DNA sequence, the more complete the DNA sequence and the higher the sequencing cost. Finally, DNA sequences are assembled through enough small fragments [10]. The primary task of sequencing data is to align the sequences and align the corresponding fragments to the DNA sequence. However, there are a large number of repeat fragments in the DNA sequence. It is estimated that the repeat length of more than 1.5% of the human genome sequence is 1000 bp The above [11], the current second-generation sequencing sequencing data length of 150pb. Traditional SNP and indel methods are based on a combination of tools such as BWA [12] + SAMTOOLS [13] + BCFTOOLS [14], BWA + PICARD + GATK [15], BWA + SAMTOOLS + Freebayes [16] The VCF file is generated at the mutation point. In this study, we did not consider filtering normal human SNP and Indel [17]. In the follow-up research, we will give full consideration to this work.

## Results

### DNA targeted sequencing data description

Targeted sequencing is a method of sequencing specific exons of a gene, which has the advantages of being targeted, reducing costs, and different sequencing targets for different cancer-targeted sequencing. For example, in the study of breast cancer high-throughput sequencing data, a total of 703 exons of 20 genes were sequenced at a depth of 1000.

DNA target sequencing data The original data needs to be cleaned before it can be processed further. The structure of the library under test and the index & R formula of the sample file are shown in Fig 1:

Fig 1 shows sequencing data in two directions of the sequencing data, there are two files R1.fq and R2.fq, respectively, in the exon region is relatively short, the two sequences will have overlapping sub-division. Index & R fine structure (Note: PE1.0 to PE2.0 direction, from 5 'to 3' end, the black sequence for the index sequence, Index structure for the 5–8 base +19 base fragment, the sample is single End index, in the index were named index & R close to PE2.0.

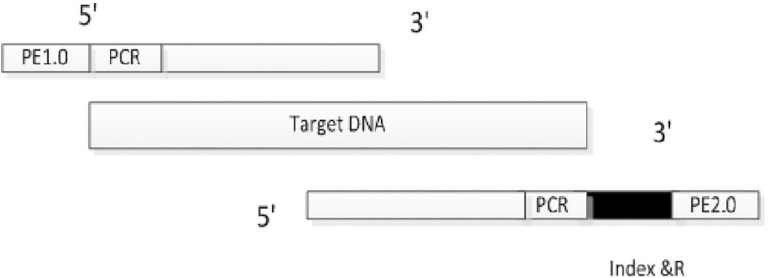

**Fig 1. DNA target sequencing original data structure.**

Which PE1.0, PE2.0 for a fixed length, you can directly clean the raw data, the raw data after cleaning only Target DNA and index data.

## PCR primer sequence introduced

As can be seen from Fig 1, the PCR primer sequences are located at both the beginning and the end of the sequencing sequence (the beginning and the end of the two-terminal sequencing). The length between the two primer sequences is less than 200, and the length of the exon region is generally designed to be less than 200 This value ensures that both sides of the sequencing overlap in the design.

The flow of the method in this article is as follows:

Step1: Data cleaning part (index cleaning and public sequence cleaning) to ensure sequencing are the target sequence;

Step2: Studying PCR primer sequences to find the optimal eigenvalue sequence and establishing quad tree data index structure;

Step3: Using the local optimal contrast algorithm sequence comparison with the target sequence to find the optimal sequence;

According to the optimal sequence structure, we calculated the SNP and InDel information of each locus.

## Public sequence cleaning

The public sequence is used to capture the DNA sequence. If the DNA sequence has not been successfully captured, the public sequence will be considered as the DNA sequence. If the capture is successful, no common sequence will appear in the original sequence. Public sequence captures DNA sequence shown in Fig 2:

As can be seen from Fig 2A, the DNA sequence has been successfully captured, and the sequence of Target DNA do not include the Public sequence, with a length of 150 bp. In Fig 2B, DNA sequences are

captured and tagged with a common sequence. If the sequence contains a common sequence, this sequence does not capture the target sequence. When double-end sequencing of the data is possible, two common sequence cross-trapping occurs, It is possible that a segment contains the target sequence and the other segment contains a common sequence, as shown in Fig 2C. Public sequence cleaning is to obtain a better sequencing sequence to improve the efficiency of alignment. For the above situation to be cleared, the principle of liquidation is as follows:

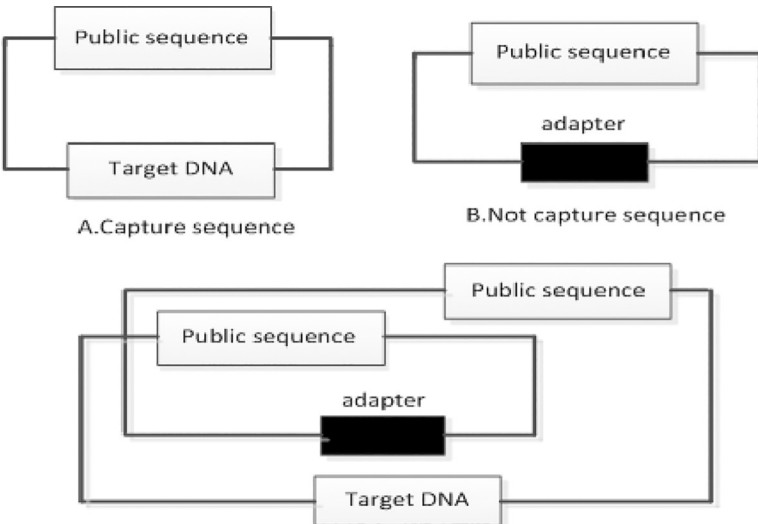

**Fig 2. DNA sequence capture.**

1. If the sequences on both ends contain a common sequence, then both ends of the sequencing data should be cleared;

2. If one end sequence contains a common sequence and the other end contains a target sequence, both ends of the sequencing data are still to be cleared;

The cleaning algorithm is as follows:

```
Algorithm 1
Input: Input the fastq file:R1,R2
Output: Read1,Read2 without the public data sequence
ReadR1 = readFastq("****R1.fq")
ReadR2 = readFastq("****R2.fq")
PubClean<-function(ReadR1,ReadR2){
Pubseq ="public data sequence"
revseq = as.character(reverseComplement(DNAString(pubseq)))
t1 = grep(pubseq,sread(ReadR1))
t2 = grep(revseq,sread(ReadR2))
sta = sort(c(setdiff(t1,t2),t2))
FLA =! sread(ReadR1) %in% sread(readt10)[sta]
read1 = sread(ReadR1)[FLA]
read2 = sread(ReadR2)[FLA]
return(Read1,Read2)}
```

## Index cleaning

Index data sequence in another reverse sequencing file for bidirectional sequencing, all cleanup at the beginning of the index sequence is taken into account, as well as the mismatch.

DNA target sequencing, the sequencing company sequencing samples based on different sequences, when the target sequence is ideal, the linker sequence will not affect the target sequence; if the target sequence is relatively short, the other sequence may be sequenced to the index, so Resulting in sequence S2 sequence data in the 3 'end coincide with index end sequence. As shown in Fig 3:

The SNP detection method mentioned in this article, you must clear this part of the sequence, or Part index data will generate more SNP points. To determine if Part Index data

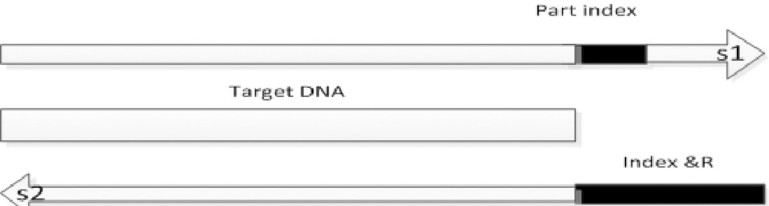

**Fig 3. Bidirectional sequence in the index part of the sequence.**

needs to be cleaned, compare the length of the Targe DNA to less than 150, and if it is less than 150, clean the excess Part Index.

Cleaning algorithm is as follows:

```
Algorithm 2
Input: Read1,Read2
Output: R1,R2 sequence data without the index sequence
Cleanindex<-function Clean(Read1,Read2){
  #Clean index in Read2 consider the mismatch situation
  for(i in 1:length(Read2)){
    pos = matchPattern(index,substr(Read2[i],1,nchar(index)+2),max.
mismatch = 3)
  if(length(width(pos))! = 0){
     Read2[i] = substr(Read2 [i],end(pos)+1,nchar(as.character(Read2
[i])))
    }
}
}
# Clean index part
  for(i in 1:length(Read1)){
   If (nchar (TargetDNA)<150){
      Read1[i] = substr(Read[i],1,nchar(TargetNDA))
   }
  }
  Return(R1,R2)}
```

**Sequencing data processing.** For sequencing, the amount of data is large, there are tens of millions of data, but there is a large number of repetitive sequences in these data. In the perfect case, the number of non-repetitive data in sequencing is equivalent to the number of target sequences. Due to system parameter settings, environmental factors, and sequencer errors, a large number of sequencing errors exist in the sequencing sequence. Therefore, in order to effectively compare sequences and reduce the number of repetitive sequences, we deal with the original data in the following way:

Step1: The original data R1, R2 data splicing to ensure that the same target sequencing Pair spliced together at both ends of the sequence;

Step2: Statistics mosaic sequence frequency appears; as shown:

Read from the Fig 4 above Read1, Read2 sequences directly connected together, due to the length of the target sequence exists Read2 and Read1, after the length of the sequence is different. Under normal circumstances, most of the target sequences are less than 200 in length, Overlap exists at the first end of the pair of target sequences, and two Overlap sequences exist in the spliced sequence. The length of the sequence after splicing varies. After statistics of the spliced sequence frequency, most sequences may have several bp differences, which is caused by the error of the sequencer.

Sequencing sequence processing algorithm is as follows:

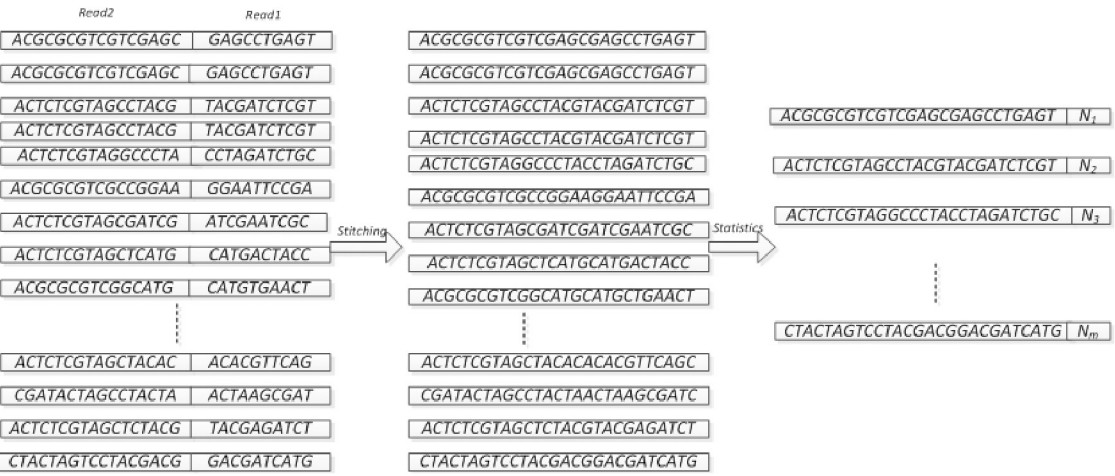

**Fig 4. Sequencing sequence processing.**

```
Algorithm 3
Input: R1, R2
Output: newtable include the R2&R2 and the frequency of sequence
newtable<-function(R1,R2){
  tablet1t2 = as.data.frame(table(paste0(as.character(R2),
    as.character(reverseComplement(R1)))))
  tablet1t2$Var1 = as.character(tablet1t2$Var1)
  t12f = order(-tablet1t2$Freq)
  tablet1t2$Freq = tablet1t2$Freq[t12f]
  $Var1 = tablet1t2$Var1[t12f]
  times = tablet1t2$Freq> = Φ
newtable = data.frame(Var1 = as.character(tablet1t2$Var1[times]),
    Freq = tablet1t2$Freq[times],stringsAsFactors = FALSE)
Return(newtable)}
```

The above code times = tablet1t2 $ Freq> = $\Phi$ means that for the frequency of occurrence is $\Phi$, when $\Phi = 1$, for all sequences to be counted, properly adjusted $\Phi$, to obtain the sequence to meet the relevant frequency, saving a lot of comparison time. The new table data frame is for the frequency greater than or equal to $\Phi$ of the sequence.

## Method

### Feature sequence extraction method

In the PCR primer sequence [18], there is a unique DNA sequence, the primer sequence is at both ends of the target sequence, and the distance between the two ends of the DNA sequence is unique. If the sequencing exon is relatively small, the sequencing of the double-ended sequencing data samples by PCR primer sequences allows for direct discovery of SNPs and InDels by matching successful sequences to targets. In order to improve the efficiency of comparison, primer sequences in this article are different in length, and the characterizes are extracted from the primer sequences, and the number of sequences can be reduced by extracting the characteristic sequences.

Characteristic sequence: the characteristic sequence between two sequences r and s (note FS (r, s)), take the same character in r, s instead of the eigenvalue.

For example, if the primer sequence R = "ACGTGC" and S = "ACGCGC" then FS (r, s) = {4: TC}, then the DNA sequence to find the one that belongs to that primer only needs to take

the fourth character of the primer sequence, If the fourth character is T, the sequencing data belongs to R; if the fourth position is C, the sequencing data belongs to S.

Generally, two sequences, if there are multiple locations, need to find many different locations can be. When multiple primer sequences are searched, eigenvalues of multiple primer sequences need to be extracted.

The extraction process is as follows:

Primer sequence $R = \{r_1, r_2, r_3, \ldots, r_n\}$, $r_i$ and $r_j$ find the smallest $FS(r_i, r_j)$, re-merge the same location information $FS(r_i, r_j)$, the final location information extracted location.

For example, R = {"ACGGGTG", "ACAGCTG", "ACCGATG", "ACCTAGG"}. There are four primer sequences in R, and in principle, {A, C, G, T} One position can distinguish these four primer sequences, $FS(r_1, r_2) = \{3{:}G\text{-}A, 5{:}T\text{-}C\}$, $FS(r_3, r_4) = \{4{:}G\text{-}T, 6{:}T\text{-}G\}$, Indicating that the same location does not exist in different characters, in this case at least two location information for standard real, so a combination of location information, CoPo = {{3,4},{3,6},{4,5}, {5,6}}, Through these four combinations respectively CoPo = {{GG, AG,CG,CT},{GT,AT,CT, CG},{GG,GC,GA,TA}, {GT,CT,AT,AG}}, Any one of the above four combination positions can represent the feature position.

Feature sequence extraction algorithm is as follows:

```
Algorithm 4
Input: R<-PCR sequence data
Output: PCR feature location information
Position<-function position(R){
length<-read(R)
For (i = 1 in 1: (length-1)){
  For(j in (i+1): length){
    FS((i-1)*(length-1)+j-i) = FS(r_i,r_j)
    }
}
Sort(FS)
Select FS In front of sever- al items: r_i∈FS_n
Position<-Select most position in FS_n # if Position< = Int
(log_4(length))
Do While (Position< = min(char(r_i))) {
If (∩_{i=1}^{length} r_i^k! = Φ k∈Position){
    Position = PositionUselect new position in FS_n
  }else{
  Break
  }
}
Return(Position)}
```

## Location-based PCR sequence comparison algorithm

The foregoing method for determining the location eigenvalue can effectively reduce the number of matching times and quickly and accurately find the location of the eigenvalue so as to make a comparison between the target sequences. The PCR sequence comparison model is shown in Fig 5 below:

The location-based PCR algorithm is as follows:

```
Algorithm 5
Input: Position,PCR, newtable
Output:Sequence with the Pcr
Com<-function(PCR,newtable,position){
  length<-length(Position)
  mathseq =""
  mathpcr =""
```

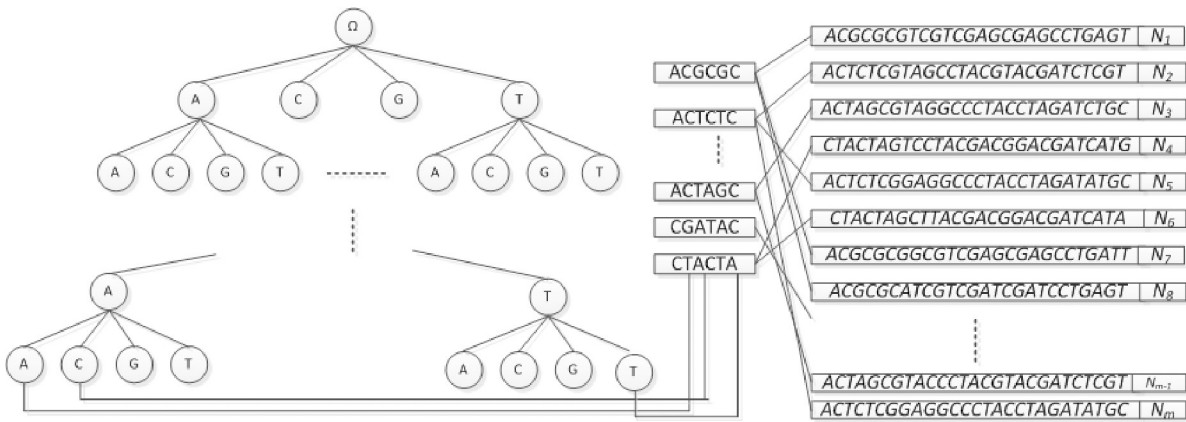

**Fig 5. PCR sequence alignment model.**

```
For (j in1: length){
   mathseq = paste0(mathseq,substr(newtable$Var1,Position[j],Posi-
tion[j])
   mathpcr = paste0(mathpcr,substr(Pcr,Position[j],Position[j]))

   math = mathpcr %in% mathseq
   Comseq = data.fram(newtable$Var1[math], newtable$Freq[math])
Return(Comseq)}
```

In Comseq, all sequences that match the Pcr signature sequence are combined, and there may be a single point variation in these signature sequences, where a single point mutation is included in the sequence.

## Local optimal solution algorithm

Let the length of the sequence S, T be m and n, respectively, using the score matrix D of data structure (m + 1) * (n + 1), the value of each element in the matrix represents the sequence 0: S: i≤m) the best alignment of a suffix with a suffix of sequence 0: T: j (0≤j≤n).

Except for the prefix score in the local alignment, the initial score matrix is as follows:

$$
\begin{aligned}
d_{0,j} = 0 \quad (0 \leq j \leq n) \\
d_{i,0} = 0 \quad (0 \leq i \leq m)
\end{aligned}
\tag{1}
$$

Since 0: s: i and 0: t: j always have an empty suffix score of 0, all elements in matrix D are greater than or equal to zero. Thus, the scoring matrix of any element is calculated as follows:

$$
d_{i,j} = \max \begin{cases} d_{i-1,j-1} + p(s_i, t_j) \\ d_{i-1,j} + p(s_i, -) \\ d_{i,j-1} + p(-, t_j) \\ 0 \end{cases}
\tag{2}
$$

In which, $p(s_i, t_j) = \begin{cases} 1 & s_i = t_j \\ 0 & s_i \neq t_j \end{cases} \quad p(s_i, -) = p(-, t_j) = -1$

A threshold of 0 means that the 0 element distribution area of the matrix corresponds to a dissimilar sequencing, while the positive number area is a locally similar area. Finally, find the maximum value in the matrix, which is the optimal local alignment score, and the

corresponding point is the unmatched point of the local alignment of the sequence. Then, the previous optimal path is reversely deduced until the local ratio The starting point.

Since the sequence determined by PCR has a great similarity with the target sequence in principle, if the similarity is not high, the sequence is deleted directly. Therefore, when comparing, only a few items need to be compared, and others need not be aligned. The calculation formula of any element in the scoring matrix is as follows:

$$
d_{i,j} = \max \begin{cases} d_{i-1,j-1} + p(s_i, t_j) & |i-j| \le \varepsilon \\ d_{i-1,j} + p(s_j, -) & |i-j| \le \varepsilon \\ d_{i,j-1} + p(-, t_j) & |i-j| \le \varepsilon \\ \qquad 0 & |i-j| > \varepsilon \end{cases} \tag{3}
$$

$\varepsilon$ is an integer, when $\varepsilon = 0$, said S is completely T sequencing, $\varepsilon = 1$ indicates that there is a position deviation. In this paper $\varepsilon = |$ m-n $| +1$ said

The local optimum contrast process is as follows:

Step1: Based on the initialized matrix D, pass the $d_{0,j} = 0$ of the first row of the score matrix and initialize the first column of the matrix $d_{i,0} = 0$.

Step2: Each element in D is calculated, and each element in the score matrix D is sequentially calculated starting from d11 according to the formula (3) and the score function formula (1) until $d_{n+1,m+1.}$

Step3: Find the optimal path to determine the maximum dij position. Reverse the forward, find the optimal path.

Step4: According to the optimal path, the optimal local alignment of bar sequences is obtained.

Example 2: Local alignment of sequences S = "CGTGAGCTG" and T = "CGTCGAGCTGA" by local alignment method, n = 11, m = 9, then $\varepsilon = 3$ The results are shown in Fig 6 below.

In the above figure, the optimal path is $d_{11,10}$-> $d_{10,9}$-> $d_{9,8}$-> $d_{8,7}$-> $d_{7,6}$-> $d_{6,5}$-> $d_{5,5}$-> $d_{4,4}$-> $d_{3,3}$-> $d_{2,2}$-> $d_{1,1}$-> $d_{0,0}$. The result is shown in Fig 6. Find the optimal local alignment from the above path, as shown in Fig 7.

According to the above scoring matrix, the optimal path is derived in the backward direction. The Algorithm 6 is as follows:

```
Algorithm 6
Input: S, T
Output: Locally optimal S, T
Com<-function(S,T){
  lengths<-nchar(S)+1
  lengtht<-nchar(T)+1
  D = matrix(0, nrow = lengtht, ncol = lengtht, byrow = T)
  (i in 2:lengtht){
  For(j in 2:lengths){
  If(|i-j|< = lengtht-lengths+1){
  If(S[i] = = T[j]){ D[i,j] = D[i-1,j-1]+1}
  Else {
      D[i,j] = max(D[i-1,j],D[i,j-1])-1}
    }
    }
  score = 0
  Do while (maxi> = 2&&maxj> = 2){
    Write(S[maxi])
```

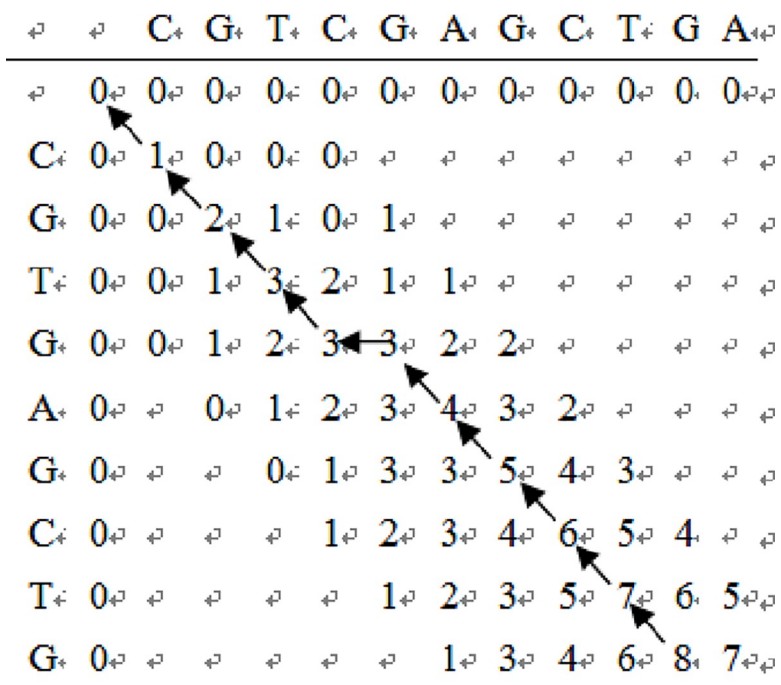

**Fig 6. Local comparison scoring matrix of S and T.**

```
   = max(D[maxi-1,maxj],D[maxi,maxj-1],D[maxi-1,maxj-1])
 If(maxd = D[maxi-1,maxj]) maxi = maxi-1
  If(maxd = D[maxi,maxj-1] maxj = maxj-1
  If(maxd = D[maxi-1,maxj-1]){
  maxi = maxi-1,maxj = maxj-1
  score = score+1}
 }return(score)
```

The above algorithm obtains the optimal path, and after determining the optimal path, returns the score. The score must satisfy the requirement and can be considered as the local optimum of S and T.

$$k = \min(lengtht, lengths) - score \tag{4}$$

K denotes the number of unmatched points between S and T. When K = 0, it indicates that S and T are complete subset relations. Generally, k is considered as 2, k is too large, and there is no local maximum between S and T. excellent. The smaller k indicates the local optimum, and the minimum is 0.

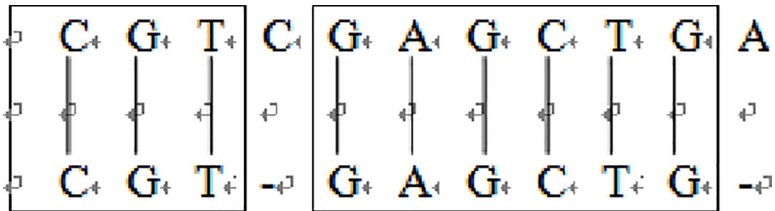

**Fig 7. Optimal local alignment of S and T.**

## SNP, Indel discovery algorithm

Based on the local optimal alignment algorithm, the local optimal method was used to find the sequence SNP, Indel. Firstly, the eigenvalues are extracted from the PCR sequence to find the feature location information, and the location data is used to search the sequencing data for the sequencing data in a specific area. After determining the sequencing data for a particular region, compare the DNA sequences for that particular region, and find the SNP, Indel information in that region. The process is shown in Fig 8 below:

Through the PCR sequence to find SNP, Indel method, the reference sequence alignment to achieve, when the reference sequence length is a certain value, there is overlap between the sequencing data at both ends, if the two ends of the sequencing data were performed with reference to the sequence For local comparisons, Indel is present if the score is different from the length of the single-end sequencing data. Indel is not present if the two are the same length, allowing for the presence of an SNP. In front of the sequence to find SNP, InDel information, because by PCR to solve the sequence is spliced through R1, R2, so to spliced the sequence split into R1, R2, and then through R1, R2 and the target sequence Local comparison. Calculate the number of A, C, G, T and N for each position of Target and calculate the SNP and InDel information finally. The analysis model is shown in Fig 9 below:

By using $P_{i1}, P_{i2}, P_{i3}$ and $P_{i4}$, if the position of i in the reference sequence is A, the ratios at i position are:

$$P(A_i) = \frac{p_{i1}}{p_{i1} + p_{i2} + p_{i3} + p_{i4} + p_{i5}} * 100\%$$

$$P(C_i) = \frac{p_{i2}}{p_{i1} + p_{i2} + p_{i3} + p_{i4} + p_{i5}} * 100\%$$

$$P(G_i) = \frac{p_{i3}}{p_{i1} + p_{i2} + p_{i3} + p_{i4} + p_{i5}} * 100\%$$  (5)

$$P(T_i) = \frac{p_{i4}}{p_{i1} + p_{i2} + p_{i3} + p_{i4} + p_{i5}} * 100\%$$

If the value of P ($A_i$) is 100%, there is no SNP at position i, and if P ($A_i$) is less than 90%, there is a variation. If the Indel phenomenon is represented by K, when k = 0, there is no Indel When K $\neq$ 0, the size of K is analyzed. If K is particularly small, it may be SNP or Del phenomenon, and the length of the sequence after sequencing becomes longer, indicating that there is a Del phenomenon, and if there is no change, an SNP phenomenon exists. If K is particularly likely to be In, there is no match for the sequencing sequence to start the match backward and backward to find the reference sequence.

## Discussion

The experimental environment adopted in this paper is Inter (R) Core (TM) I7-6700, memory 8G, and R language. The comparison targets are BWA + SAMTOOLS + BCFTOOLS (BSB), BWA + PICARD + GATK (BPG), BWA + SAMTOOLS + Freebayes (BSF). Three main methods of finding SNPs are compared with those presented in this paper based on PCR methods.

Using Wgsim software [19] in this study, the software is better than other software in terms of controllability, which can limit the error rate of sequencing, sequencing length, and sequencing type. Today's high-throughput sequencing market is dominated by Illumina's double-ended sequencing technology [20], and Wgsim's mock-sequencing software is well-suited for double-ended mimic sequencing. Pre-filtration through the sequence of low-quality sequence of cleaning, cleaning effect as shown in Table 1 below, through the control of low

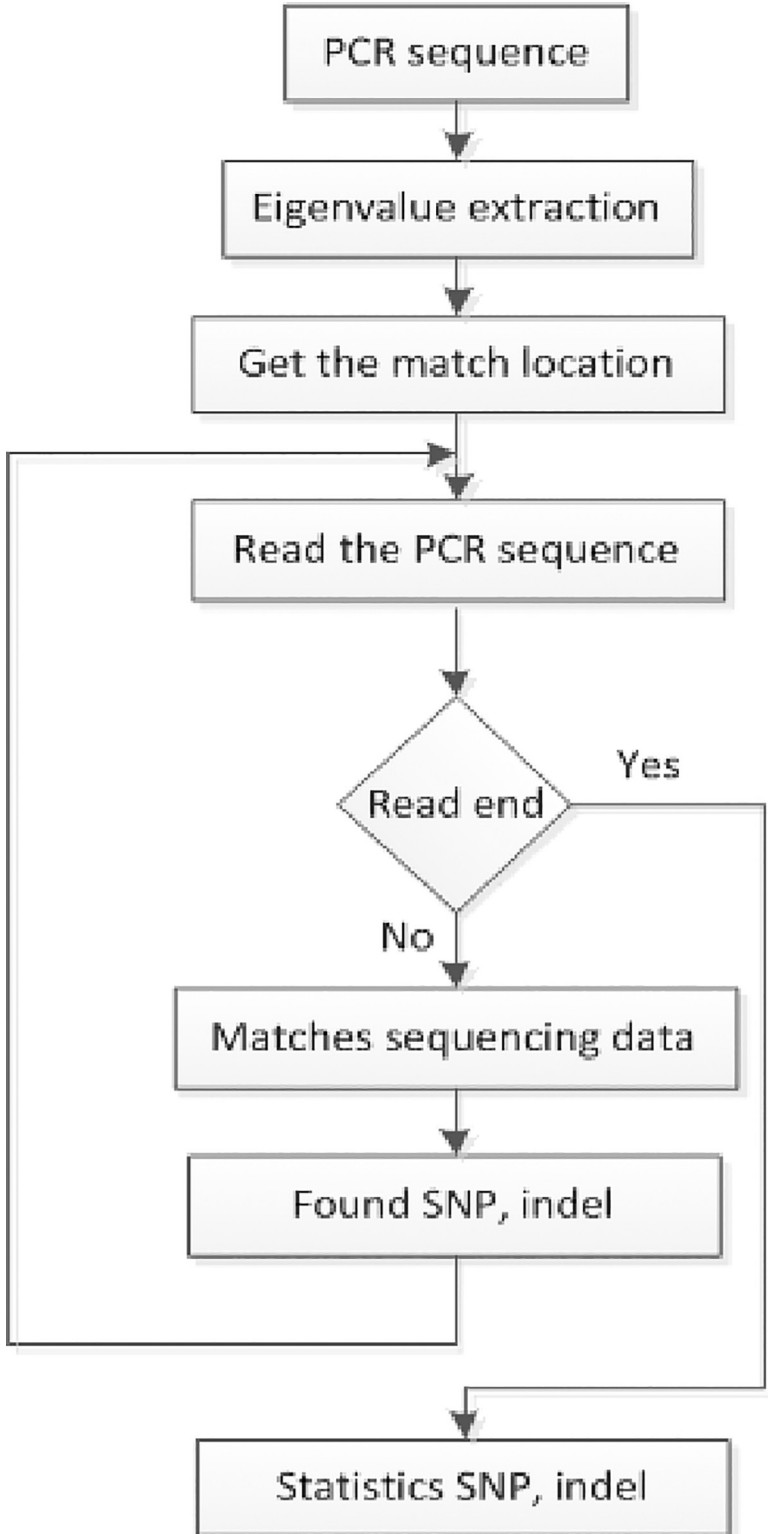

**Fig 8. SNP and Indel discovery process.**

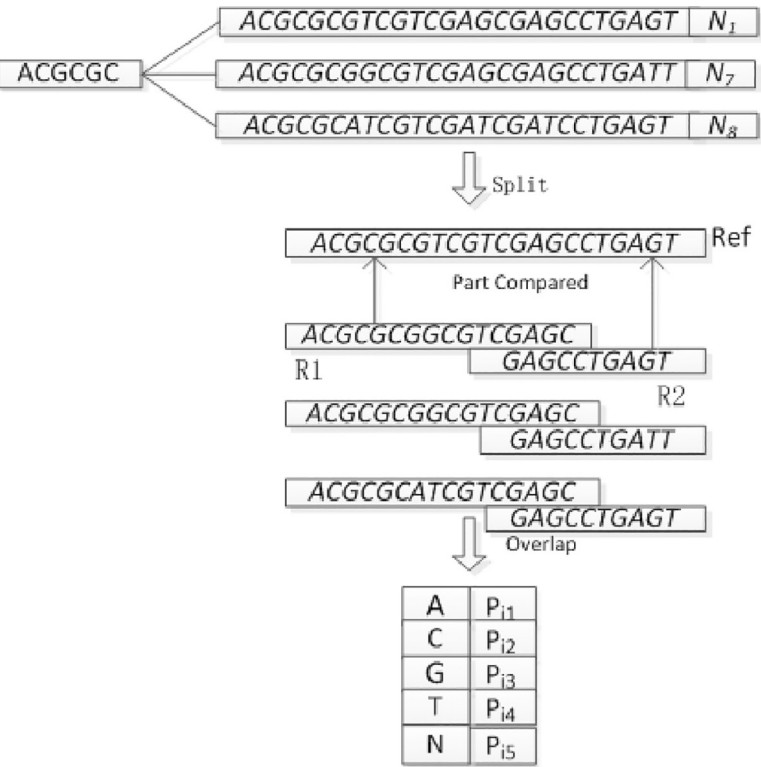

**Fig 9. Local alignment SNP analysis model.**

quality, clean-up before and after the effect, we can see that after the treatment of the average mass ratio increased.

The sequencing data was sequenced at a depth of 50, 100, 150, and a length of 50, 100, 150. With the increase of sequencing depth, the accuracy of sequencing has been improved to a certain extent. When the length of sequencing data is 150, the effect is the best, and the correct rate can be achieved. Mainly due to the increased sequencing depth, the number of sequencing in the same section increased, the accuracy rate increased, and if the length of the sequencing data was increased, the correct rate in the DNA sequence was also continuously improved.

**Table 1. Data comparison before and after cleaning data.**

| Sample | Deep | Raw_len | Raw_reads | Clean_reads | rate |
|--------|------|---------|-----------|-------------|------|
| T1 | 50 | 50 | 100221025 | 91652127 | 91.45% |
| T2 | 50 | 100 | 100354125 | 91553068 | 91.23% |
| T3 | 50 | 150 | 105145621 | 97417418 | 92.65% |
| T4 | 100 | 50 | 232154587 | 216182351 | 93.12% |
| T5 | 100 | 100 | 223651478 | 209427244 | 93.64% |
| T6 | 100 | 150 | 236548754 | 217979677 | 92.15% |
| T7 | 150 | 50 | 325412548 | 304488521 | 93.57% |
| T8 | 150 | 100 | 326548456 | 307706610 | 94.23% |
| T9 | 150 | 150 | 327845123 | 309321874 | 94.35% |
| T10 | 200 | 50 | 425896544 | 398000320 | 93.45% |
| T11 | 200 | 100 | 468542158 | 440991879 | 94.12% |
| T12 | 200 | 150 | 487459562 | 462842854 | 94.95% |

Table 2. SNP and Indel comparison (depth = 50).

| Soft | Snp | Indel | Snp_rate | Indel_rate |
|------|-----|-------|----------|------------|
| PCR | 3421 | 1925 | 97.74% | 96.25% |
| BSB | 3054 | 256 | 87.26% | 12.80% |
| BPG | 3153 | 1715 | 90.09% | 85.75% |
| BSF | 2985 | 1636 | 85.29% | 81.80% |

SVsim tools [21] were used to randomly generate sequences of insertions, deletions, repeats, inversions, and translocations in the sequence to generate sequencing data, and the selected sequencing depth was 50, 100, 150. The above several software tests, from the software to see the actual situation of various software, statistical analysis of SNP and Indel, as shown in Tables 2–4:

In the analysis of Indel for comparison, due to stools in the real independence Indel, the effect is relatively poor. With the continuous increase of testing depth, the software identifies SNP. Indel effect is obviously improved. Increasing the depth will increase the number of comparisons, and can obviously increase the number of regional sequences and improve the recognition rate.

The above data does not filter the SNP, so find a lot of SNP points, this difference is set by the tool's own algorithm, and data differences also lead to the occurrence of this result. As shown in Figs 10 and 11, there are differences between SNP and Indel in the software. Because there is a small number of BSF in Indel, this article does not participate in the comparison.

In the process of generating simulated sequencing data, we analyzed the BAM files with the above four kinds of software to study the recall of the four kinds of software to estimate the sensitivity of the mutation sites.

$$Recall = \frac{TP}{TP + FN} \tag{6}$$

Three thousand SNP sites were inserted into the sequencing sequence, with 2000 Indel sites (2–10 bp deletion and 2–10 bp increase). The 25 sets of sequencing data were simulated by ILLUMINA company. The sequencing fragment length was 150 bp, the standard error of sequencing was 0, and the sequencing error rate was 0. That is to say, only 300bp insertions 1bp, in this way, four kinds of analysis software (PCR, BSB, BPG, and BSF) can be used to detect the recall of simulated tumor data with deletion mutation by the single-factor controlled variable method. The following values are mean values of the test samples. The effect is shown in Figs 12 and 13.

As can be seen from the figure above, as the depth of sequencing is found to increase in terms of SNP and Indel, it indicates that there is sufficient sequencing depth in sequencing data to ensure SNP and Indel correctness. In the sequencing process can have enough sequencing depth, can have high accuracy.

Table 3. SNP and Indel comparison (depth = 100).

| Soft | Snp | Indel | Snp_rate | Indel_rate |
|------|-----|-------|----------|------------|
| PCR | 3452 | 1932 | 98.63% | 96.60% |
| BSB | 3235 | 320 | 92.43% | 16.00% |
| BPG | 3254 | 1825 | 92.97% | 91.25% |
| BSF | 3021 | 1734 | 86.31% | 86.70% |

**Table 4. SNP and Indel comparison (depth = 150).**

| Soft | Snp | Indel | Snp_rate | Indel_rate |
|---|---|---|---|---|
| PCR | 3468 | 1948 | 99.09% | 97.40% |
| BSB | 3335 | 354 | 95.29% | 17.70% |
| BPG | 3254 | 1895 | 92.97% | 94.75% |
| BSF | 3210 | 1812 | 91.71% | 90.60% |

In order to better express the effect of PCR detection, true positive rate, false positive rate and accuracy rate were used to evaluate. The equation is as follows:

$$TPR = \frac{TP}{TP + FN} \tag{7}$$

$$FPR = \frac{FP}{FP + TN} \tag{8}$$

$$Accuracy = \frac{TP + TN}{TP + FP + FN + TN} \tag{9}$$

The effectiveness of the four algorithms was evaluated by the true positive rate, the false positive rate and the accuracy rate, and the classification performance under different sequencing depths was shown in Tables 5–7.

Comparison of the performance of sequencing data in experiment When using PCR, BSB, BPG and BSF to predict, the accuracy of the original dataset processed by sampling technique was good. When depth = 150, the accuracy of PCR was 97.76%, BSB was 65.54%, BPG was 93.65% and BSF was 88.53%. The reason of low accuracy of BSB may be that the effect of mailing InDel is not ideal and the accuracy of BSB is low. As that sequence depth increases, the

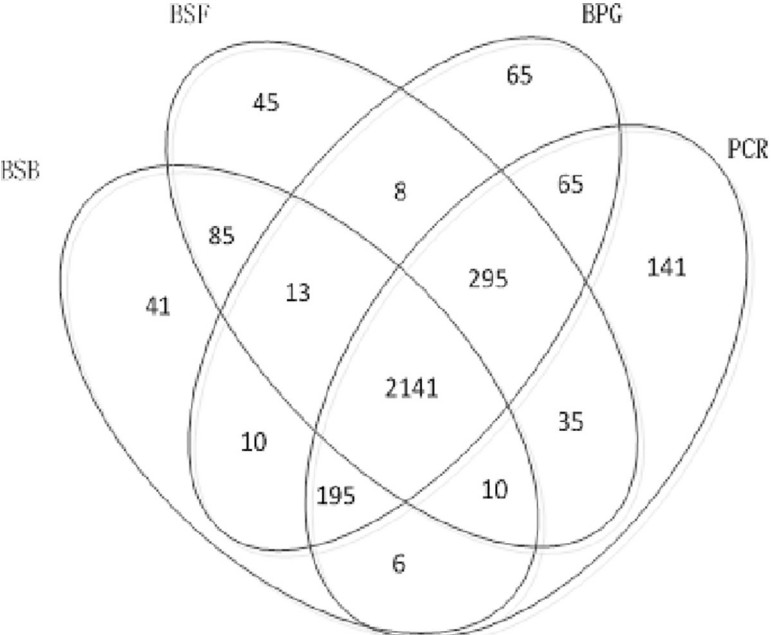

**Fig 10. Four software SNP comparison.**

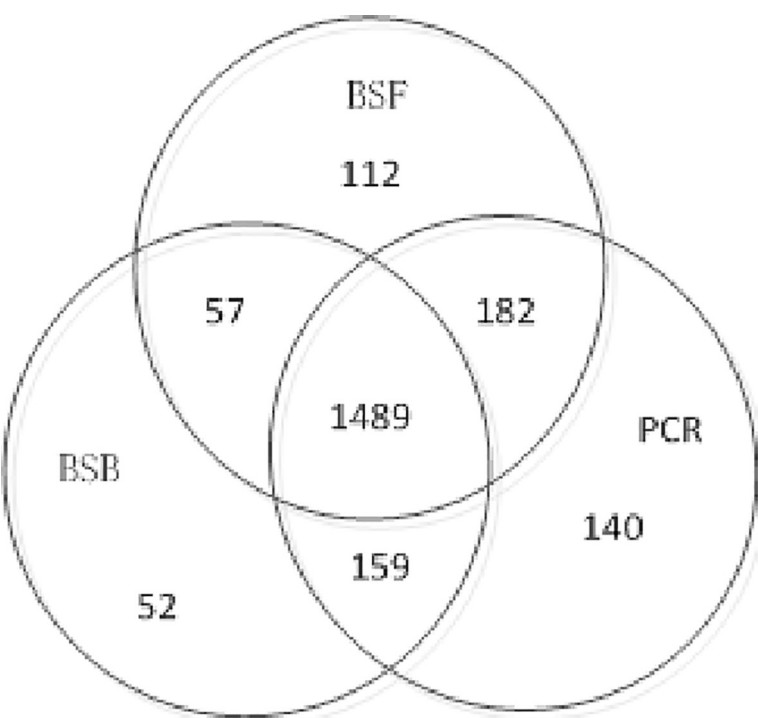

**Fig 11. Three software Indel comparison.**

PCR algorithm achieve the highest level of accuracy, but it can be seen that the accuracy of the other three methods is also rising. The increase of sequencing length can improve the efficiency of BWA alignment, and the accuracy of BPG is also increased significantly, which is also close to PCR method. Similarly, there is a similar situation in the TPR and FPR indicators, which is not explained here.

In the run time, BSB, BPG, BSF need BWA sequence alignment in the initial run time, which will take up a lot of time, while PCR directly sequencing and quantitative statistics of the initial sequence, which takes up less time. The running time of BPG is longer than that of

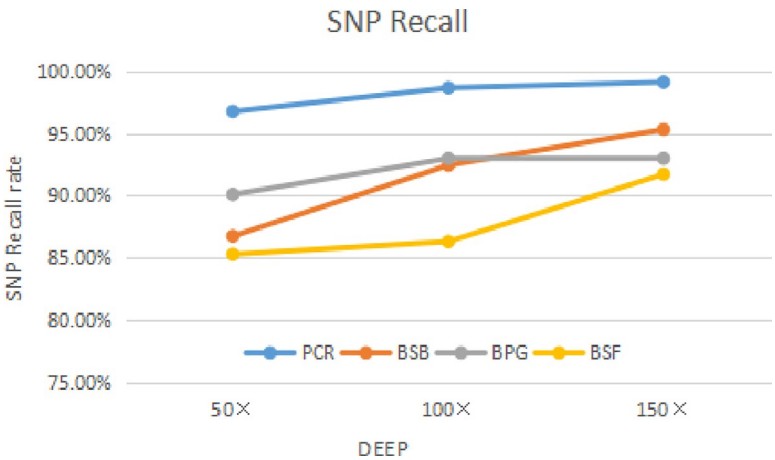

**Fig 12. Four kinds of software SNP recall rate.**

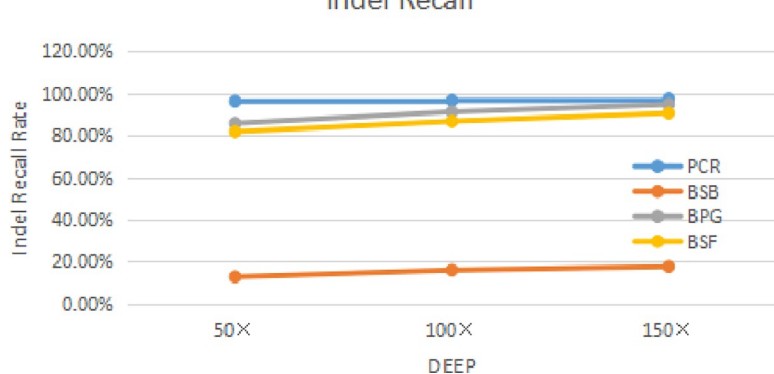

**Fig 13. Four kinds of software Indel recall rate.**

**Table 5. Classification of four algorithms (depth = 50).**

| Soft | TPR | FPR | Accuracy | Time(s) |
|------|-----|-----|----------|---------|
| PCR | 95.65% | 68.76% | 96.84% | 57 |
| BSB | 65.79% | 85.54% | 58.54% | 159 |
| BPG | 87.76% | 78.65% | 88.43% | 224 |
| BSF | 79.49% | 82.65% | 82.76% | 187 |

**Table 6. Classification of four algorithms (depth = 100).**

| Soft | TPR | FPR | Accuracy | time |
|------|-----|-----|----------|------|
| PCR | 96.12% | 62.34% | 97.13% | 74 |
| BSB | 67.79% | 83.74% | 61.54% | 187 |
| BPG | 91.23% | 76.65% | 91.65% | 254 |
| BSF | 81.58v | 80.65% | 86.23% | 201 |

**Table 7. Classification of four algorithms (depth = 150).**

| Soft | TPR | FPR | Accuracy | time |
|------|-----|-----|----------|------|
| PCR | 97.25% | 54.76% | 97.76% | 96 |
| BSB | 71.24% | 80.74% | 65.54% | 214 |
| BPG | 94.76% | 74.54% | 93.65% | 267 |
| BSF | 83.87% | 78.27% | 88.53% | 227 |

GATK except in BWA. But the effect of GATK is also ideal. As that length of the sequence increase, the running time of PCR decrease, and as the length of the sequence increases, the comparison time also increases, so the running time also increase.

## Conclusions

With the development of sequencing technology, sequence can be analyzed directly from exon sequencing, and SNP and InDel in sequence can be obtained quickly, which is the

fundamental goal of sequencing. In this paper, PCR primers are included in sequencing sequence, and the comparison between PCR primer sequence and sequencing data can quickly achieve this goal. In this way, the sequencing data can also be analyzed under the general condition of computer hardware. Compared with other traditional methods, the performance of the proposed method is much better, and the accuracy and time of the proposed method are also very high. In the future, we will focus on how to obtain the sequence of mutation quickly under the condition of specific gene mutation. Especially, SNP, InDel or CNV mutate specific sequences, and use PCR primers or set specific sequences to find the mutated sequences in the sequence data.

## Acknowledgments

We thank the relevant researchers from Chongqing Cancer Hospital for their guidance of this paper.

## Author Contributions

**Writing – review & editing:** Guobin Chen, Xianzhong Xie.

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
