## [Decision Letter · Decision Letter 0]

7 Apr 2020

PONE-D-19-33405

An Exon Sequencing Mutation Detection Algorithm Based on PCR Matching

PLOS ONE

Dear Dr Xie,

Thank you for submitting your manuscript to PLOS ONE. After careful consideration, we feel that it has merit but does not fully meet PLOS ONE’s publication criteria as it currently stands. Therefore, we invite you to submit a revised version of the manuscript that addresses the points raised during the review process.

We would appreciate receiving your revised manuscript by May 22 2020 11:59PM. To enhance the reproducibility of your results, we recommend that if applicable you deposit your laboratory protocols in protocols.io, where a protocol can be assigned its own identifier (DOI) such that it can be cited independently in the future. For instructions see: http://journals.plos.org/plosone/s/submission-guidelines#loc-laboratory-protocols

We look forward to receiving your revised manuscript.

Kind regards,

Jumana Yousuf Al-Aama, MD, SBP, MRCP, FCCMG

Academic Editor

PLOS ONE

Journal Requirements:

2. PLOS ONE has specific requirements for studies that are presenting a new method as the primary focus, which we believe is the case for your manuscript. Specifically, we require these submissions to meet the criteria of utility, validation, and availability (https://journals.plos.org/plosone/s/submission-guidelines#loc-methods-software-databases-and-tools). In light of this, please include a copy of the open-source code for your algorithm as a Supporting Information file or provide a link if it is available through an online repository. For more information of PLOS ONE data availability, please see https://journals.plos.org/plosone/s/data-availability

Reviewers' comments:

Reviewer's Responses to Questions

**Comments to the Author**

1. Is the manuscript technically sound, and do the data support the conclusions?

Reviewer #1: Yes

Reviewer #2: Partly

2. Has the statistical analysis been performed appropriately and rigorously? 

Reviewer #1: Yes

Reviewer #2: Yes

3. Have the authors made all data underlying the findings in their manuscript fully available?

Reviewer #1: Yes

Reviewer #2: Yes

4. Is the manuscript presented in an intelligible fashion and written in standard English?

Reviewer #1: Yes

Reviewer #2: No

5. Review Comments to the Author

Reviewer #1: Reviewer Recommendation and Comments for Manuscript Number PONE-D-19-33405

• An article in Nature Scientific reports (Ref given) has specified about primer-induced nucleotide labeling for massive sequencing on next-generation sequencing platforms by primer-induced sample labeling method for sequencing a large number of samples simultaneously on NGS. This article has specified on use of nucleotide labelling but the current article gives an insight into use of PCR primers for sequencing and with supporting algorithms. This is a novel idea and is applicable in this next gen sequencing era.

• The manuscript is well written and researched.

• The manuscript is acceptable to be published in PLoS One.

Ref: Guo, J., Cheng, T., Xu, H. et al. An efficient and cost-effective method for primer-induced nucleotide labeling for massive sequencing on next-generation sequencing platforms. Sci Rep 9, 3125 (2019). https://doi.org/10.1038/s41598-019-38996-8

Reviewer #2: The overall language used and the format of the article needs to be changed to better suit this Journal.

A bigger pool of test sample should be used to validate the data with negative controls as well.

6. PLOS authors have the option to publish the peer review history of their article (what does this mean?). If published, this will include your full peer review and any attached files.

Reviewer #1: No

Reviewer #2: No

---

## [Author Response · Author response to Decision Letter 0]

26 May 2020

Reviewer Recommendation and Comments for Manuscript Number PONE-D-19-33405

• An article in Nature Scientific reports (Ref given) has specified about primer-induced nucleotide labeling for massive sequencing on next-generation sequencing platforms by primer-induced sample labeling method for sequencing a large number of samples simultaneously on NGS. This article has specified on use of nucleotide labelling but the current article gives an insight into use of PCR primers for sequencing and with supporting algorithms. This is a novel idea and is applicable in this next gen sequencing era.

• The manuscript is well written and researched.

• The manuscript is acceptable to be published in PLoS One.

Ref: Guo, J., Cheng, T., Xu, H. et al. An efficient and cost-effective method for primer-induced nucleotide labeling for massive sequencing on next-generation sequencing platforms. Sci Rep 9, 3125 (2019). https://doi.org/10.1038/s41598-019-38996-8

Answer: Add the reference to the paper.

Reviewer #2: The overall language used and the format of the article needs to be changed to better suit this Journal.

Answer: Your suggestion is kindly appreciated. I have looked through the overall manuscript and made corresponding revision grammatically to make more smooth reading.

A bigger pool of test sample should be used to validate the data with negative controls as well.

Answer: Thanks for your kind suggestion. Owing to the Covid-19, I failed to retrieve some sample data from the experiments before. What’s more, the amount of data used in this paper has already supported the practical application effect of the algorithm proposed in this paper. Thanks for your understanding,

Answer: b) is my option. The article provides a trial of data acquisition; please refer to Availability of data and material in the article for details.

---

## [Editor Report · Decision Letter 1]

14 Jul 2020

Exon Sequencing Mutation Detection Algorithm Based on PCR Matching

PONE-D-19-33405R1

Dear Dr. Xie,

We’re pleased to inform you that your manuscript has been judged scientifically suitable for publication and will be formally accepted for publication once it meets all outstanding technical requirements.

Kind regards,

Jumana Yousuf Al-Aama, MD, SBP, MRCP, FCCMG

Academic Editor

PLOS ONE